



# Calibration of Hydroxyacetonitrile (HOCH₂CN) and Methyl isocyanate (CH₃NCO) Isomers using I⁻ Chemical Ionization Mass Spectrometry (CIMS)

Zachary Finewax[1, 2,a], Aparajeo Chattopadhyay[1,2,b], J. Andrew Neuman[1,2], James M. Roberts[1], James B. Burkholder[1,*]

[1] Chemical Sciences Laboratory, National Oceanic and Atmospheric Administration (NOAA), Boulder, CO, USA
[2] Cooperative Institute for Research in Environmental Sciences (CIRES), University of Colorado, Boulder, CO, USA
[a] Present address: Air Pollution Control Division, Colorado Department of Health and Environment, Denver, CO, USA
[b] Present address: Department of Earth and Environmental Sciences, University of Rochester, Rochester, NY, USA

*Correspondence to*: James B. Burkholder (James.B.Burkholder@noaa.gov)

**Abstract.** The toxic reduced nitrogen compound methyl isocyanate (CH₃NCO, MIC) has been reported present in wildfire and bio-mass burning emissions, agricultural fumigation plumes, and indoor air. Its isomer, hydroxyacetonitrile (HOCH₂CN, glycolonitrile, HAN) has not been observed in the Earth's atmosphere to date. In this study, absolute sensitivity calibrations for these isomers using two I⁻ chemical ionization mass spectrometry (I-CIMS) instruments, time-of-flight (ToF) and quadrupole (Quad) instruments, commonly used in laboratory and field measurements, were performed, for the first time, over a range of ion-molecule reactor temperature (10–40°C) and I(H₂O)⁻/I⁻ ratio (0.01–1). This study demonstrates that I-CIMS, under typical operating conditions, is not sensitive to MIC with limits of detection (LOD) of >860 and >570 ppb for the ToF and Quad I-CIMS instruments, respectively. Both I-CIMS instruments are, however, highly sensitive to the HAN isomer with 0.3 and 3 ppt LODs for the ToF-CIMS and Quad-CIMS instruments, respectively. These results contradict several recent field studies that have reported the detection of MIC using I-CIMS instruments. This study demonstrates that HAN, rather than MIC, was most likely the C₂H₃NO isomer observed in those field studies, although the source chemistry for HAN remains uncharacterized.



## 1 Introduction

The identification and quantitative measurement of atmospheric trace species are critical elements in the development of air quality models and the establishment of environmental policies. Methyl isocyanate (CH₃NCO, MIC) and hydroxyacetonitrile (HOCH₂CN, glycolonitrile, HAN) (Wolfsie, 1960) are toxic reduced nitrogen isomers present in an extreme range of environments. MIC has been reported in wildfire and bio-mass burning, e.g., Koss et al. (2018), and building fire emissions (Blomqvist et al., 2003), agricultural fumigation plumes (Woodrow et al., 2014), and indoor air (Bekki et al., 2018), particularly where there is cigarette smoke (Moldoveanu, 2010), cooking emissions, and chlorine-disinfectant use. HAN has been observed spectroscopically in interstellar space (see Zhao et al. (2021) and references within), while measurements in the Earth's atmosphere have not been reported to date. The atmospheric degradation of MIC leads to the formation of isocyanic acid (HNCO) (Papanastasiou et al., 2020), another toxic reduced nitrogen compound. The atmospheric chemistry of the HAN isomer is not presently characterized.

The implementation of I⁻ chemical ionization mass spectrometry (I-CIMS) instruments, in particular high-resolution time-of-flight mass spectrometer instruments (I-CIMS ToF-MS), has enabled high sensitivity laboratory and ground-based field measurements as well as high spatial and temporal resolution in airborne field studies. However, a general limitation of CIMS measurements is the inability to distinguish isomers, e.g., MIC and HAN C₂H₃NO isomers, without using pre-separation techniques such as gas chromatography. Field measurements of MIC have been reported in outdoor (Priestley et al., 2018) and indoor (Mattila et al., 2020a; Mattila et al., 2020b; Wang et al., 2022) environments using (I-CIMS) instruments, with sensitivity calibration estimated using voltage-scan techniques (Mattila et al., 2020b; Wang et al., 2022) and analogy to the sensitivity of other VOC species (Priestley et al., 2018).

The primary objective of this work was to establish sample handling and I-CIMS sensitivity calibration protocols for the C₂H₃NO MIC and HAN isomers. Absolute pressure and FTIR measurements were used to establish MIC standards, while diffusion and infusion methods, coupled with total reactive nitrogen (N$_r$) methods, were used for the quantification of HAN standards. Calibration measurements are presented for I-CIMS time-of-flight (ToF-CIMS) and quadrupole CIMS (Quad-CIMS) mass spectrometry instruments over a range of I(H₂O)⁻/I⁻ ratios and ion-molecule reactor (IMR) temperatures. The results from this work can be applied to similar instruments used in laboratory and field studies, although instrument sensitivities may vary somewhat depending on instrument configuration and operating conditions. This work demonstrates that the recent field measurements reported by Priestley et al. (2018), Mattila et al. (2020a), and Wang et al. (2022) have mis-attributed their measured I-CIMS signal, which was assigned to MIC, when it was most likely due to HAN.

## 2 Experimental Methods

A primary goal of this work was the development of measurement protocols for hydroxyacetontrile (HOCH₂CN, HAN) and methyl isocyanate (CH₃NCO, MIC) using iodide chemical ionization mass spectrometry (I-CIMS) with a time-of-flight mass spectrometer (ToF-CIMS) (Lee et al., 2014) and quadrupole mass spectrometer (Quad-CIMS) (Neuman et al., 2000) instruments. We have focused on calibration measurements for ToF-CIMS and Quad-CIMS instruments for which calibrations for MIC and HAN have not been performed previously. An emphasis in this work was placed on MIC and HAN sample handling and quantification, i.e., knowing what the sample concentration is at the instrument inlet.



71       MIC and HAN samples for calibration measurements were prepared using dilute gas mixtures prepared using

absolute pressure measurements, diffusion sources, and infusion sources that are described in detail below. Absolute
concentrations were determined using FTIR spectroscopy (MIC) and a total reactive nitrogen ($N_r$) instrument (HAN).
Calibration measurements were performed with ToF- and Quad-CIMS instruments to evaluate differences in
sensitivity for instruments with different configurations, e.g., ion-molecule reactor (IMR) geometries, ionization
sources, and ion focusing optics. In addition, sensitivities were determined for a range of IMR temperature (10–40°C)
and $I(H_2O)^-/I^-$ ratios, commonly used in laboratory and field studies, which are known to affect I-CIMS sensitivity
(see Robinson et al. (2022)). The MIC and HAN calibration methods, $N_r$ instrument and methods, and ToF- and
Quad-CIMS instruments are described below.

### 2.1 Methyl isocyanate source and calibration

Methyl isocyanate (MIC) is a stable volatile liquid at room temperature with a vapor pressure of ~467 hPa (1 hPa =
0.75 Torr) at 25°C. For the present study, a MIC sample was obtained commercially in pure form. Dilute gas-phase
methyl isocyanate (MIC) samples were prepared manometrically at a total pressure of 1066 hPa in a 12 L Pyrex bulb
with He bath gas. The MIC mixing ratio was ~3.5%. The MIC mixing ratio was also determined by Fourier transform
infrared spectroscopy using the absorption cross section data measured in Papanastasiou et al. (2020) from this
laboratory. Dilute high-pressure gas standards were prepared by diluting glass bulb standards into aluminum cylinders
and filled with zero air. Methyl isocyanate standards were quantitatively added to the calibrated zero air flow sampled
by the I-CIMS instruments.

### 2.2 Hydroxyacetonitrile (HAN) source and calibration methods

HAN is a stable semi-volatile compound, vapor pressure of 1.33 hPa at 336 K (NIOSH), that is available commercially
as a 70 wt% mixture with $H_2O$. Over the course of this study, liquid-to-gas-phase diffusion and infusion methods
were applied for the delivery of gas-phase HAN to the $N_r$ calibration and I-CIMS instruments.
The diffusion source is described in detail elsewhere (Roberts et al., 2010; Williams et al., 2000). Basically, a
capillary diffusion cell regulated vapor from the liquid sample into the gas stream feeding into the $N_r$ and I-CIMS
instruments. The concentration of source compound is regulated by varying the total gas flow, i.e., dilution factor.
This source method requires an independent determination of the compound concentration in the gas stream. The
infusion method has been used previously in our laboratory and is described in detail elsewhere (Bernard et al., 2017;
Bernard et al., 2018). Basically, a syringe pump is used to deliver a constant liquid flow (0.01–0.3 µL min$^{-1}$ in our
experiments) of a HAN sample into the gas stream feeding into the $N_r$ and I-CIMS instruments. The region of the
liquid-gas interface was heated (40 to 60°C) to ensure uniform volatilization and minimize potential sample
condensation. Measurements were performed using the commercially available stock HAN solution and a HAN
sample diluted in acetone, prepared off-line. The mixing ratio of HAN in the gas flow can in principle be calculated
using the calibrated gas and liquid flow rates, the density of the compound, and its liquid mixing ratio, although in
this work, the HAN infusion source was calibrated using a total reactive nitrogen, $N_r$, measurement.

### 2.3 Total reactive nitrogen, $N_r$, measurement

The infusion source gas-phase concentration of HAN was determined by a total reactive nitrogen, $N_r$, measurement.
Total reactive nitrogen is defined as all reduced and oxidized nitrogen-containing compounds with the exception of





$N_2$ and $N_2O$. The $N_r$ instrument has been demonstrated for both gas-phase and particle-phase $N_r$, and is described in
detail elsewhere (Stockwell et al., 2018). In this study, the nitrogen containing compounds in a gas-phase sample are
first catalytically converted on a 750°C Pt catalyst to NO and $NO_2$. The $NO_2$ is subsequently converted to NO on a
molybdenum oxide catalyst at 350°C. The NO then reacts with an excess of $O_3$ to form $NO_2$, which is detected by
chemiluminescence (Williams et al., 1998). The $N_r$ instrument was calibrated using commercial dilute mixtures of
NO (5.18 ppm) and HCN (9.5 ppm) in a $N_2$ bath gas. HCN and NO calibrations performed over the course of the
study agreed to within 3%. The total flow through the $N_r$ instrument was set to 1.048 sLPM (standard liter per minute)
and the total zero air flow from the infusion source was 2.148 sLPM. The excess flow passed through an exhaust line.
Considering uncertainties in the standards and flow rates, the $2\sigma$ uncertainty in the $N_r$ calibration of the HAN source
was estimated to be 15%. The HAN infusion source was calibrated by measuring the $N_r$ concentration as a function
of the liquid injection flow rate. Over the course of the study, multiple calibration experiments were performed using
independently prepared HAN samples.
**2.4 Iodide Chemical Ionization Mass Spectrometry (I-CIMS)**
I-CIMS has the ability to measure sub-part per trillion (ppt) gas-phase concentrations of organic acids, halogens,
oxidized organic compounds, and $N_2O_5$ at up to 10 Hz resolution (Huey, 2007; Neuman et al., 2000; Veres et al.,
2020). This relatively soft ionization technique usually yields an $I^-$ cluster ion with the intact analyte molecule. I-
CIMS is highly selective, with the sensitivity to an analyte dependent on the binding enthalpy of the compound with
$I^-$. Time-of-flight mass spectrometers typically contain ion focusing quadrupoles that can also impart changes to the
sensitivity of analytes due to mass-dependent ion transmission and collision-induced dissociation (Robinson et al.,
2022). The I-CIMS instruments used in the present study have been described in detail previously and only pertinent
details are described below.
**2.4.1 Time-of-Flight Chemical Ionization Mass Spectrometer (ToF-CIMS)**
The ToF-CIMS is setup with a pressure-controlled inlet, ion molecule reactor (IMR), small segmented quadrupole
(SSQ), big segmented quadrupole (BSQ), and time-of-flight (ToF) mass analyzer (Veres et al., 2020). A Kr lamp
provided vacuum ultraviolet radiation at 124 and 117 nm that photoionized $CH_3I$ to produce $I^-$ in the IMR
(Breitenlechner et al., 2022; Ji et al., 2020). Analytes (A) react with $I^-$ or $I(H_2O)^-$ to form adducts:
$I^- + A \quad\quad \leftrightarrow [I\text{-}A]^-$         (1)
$[I\text{-}H_2O]^- + A \quad \leftrightarrow [I\text{-}A]^- + H_2O$         (2)
with $I\text{-}C_2H_3NO^-$ detected at m/z 183.9265. Key instrument conditions that impact the sensitivity for an analyte are the
IMR temperature and $I(H_2O)^-/I^-$ ratio (Lee et al., 2014; Robinson et al., 2022; Veres et al., 2020). In this study, the
IMR was temperature controlled between 20 and 40°C and the $I(H_2O)^-/I^-$ ratio was dynamically controlled at 0.01,
i.e., dry, and over the range 0.4 to 0.6, which represents a typical range of operating conditions of laboratory and field
instruments. The ToF-CIMS inlet was pressure controlled at 130 hPa and the flow into the inlet was controlled to 6
sLPM. The IMR was pressure controlled to 44 hPa (Zhang and Zhang, 2021). The pressure in the SSQ and BSQ
were 1.7 and 0.013 hPa, respectively. The sum of $I^- + I(H_2O)^-$ was typically 5MHz in these experiments.
**2.4.2 Quadrupole Chemical Ionization Mass Spectrometer (Quad-CIMS)**





The Quad-CIMS instrument setup consisted of a critical orifice inlet (600 μm dia.) combined with an IMR, ion
focusing lenses, and a quadrupole mass analyzer (Neuman et al., 2000). I$^-$ ions were generated by passing a 0.1%
CH$_3$I in N$_2$ mixture through a $^{210}$Po radioactive source. The IMR was temperature controlled between 10 and 30°C.
The I(H$_2$O)$^-$/I$^-$ ratio was adjusted between 0.01–1.0 by controlling the humidity of the gas flow into the CIMS, which
ranged from 4.2 to 4.5 sLPM. This system did not have an intermediate pressure zone between the IMR and mass
spectrometer chamber, common to many I-CIMS instruments, that would serve as a collisional-dissociation chamber.
The pressure in the IMR was maintained at 37 hPa by varying the pumping speed. The mass resolution of the Quad-
CIMS instrument is ~200 and that of the ToF-CIMS is ~5000. The sum of I$^-$ + I(H$_2$O)$^-$ was typically 500 kHz in these
experiments.
**2.5 Materials**
Synthetic air (zero grade), N$_2$ (UHP, 99.999%), He (UHP, 99.999%) gases and CH$_3$I (99%) and acetone (99%) were
used as provided. Standard dilute mixtures of NO (5.18 ppm in N$_2$) and HCN (9.5 ppm in N$_2$) were obtained
commercially. Hydroxyacetonitrile (HAN, ~70% in H$_2$O, CAS RN: 107-16-4) and methyl isocyanate (MIC, 97+%,
CAS RN: 624-83-9) samples were obtained commercially. The MIC sample contained a <3% trimethylchlorosilane
(CAS RN: 75-77-4) inhibitor. The HAN and MIC samples were degassed in freeze—pump—thaw cycles. Samples
were stored in a chemical refrigerator in vacuum sealed Pyrex reservoirs prior to use.
A dilute (3.48%) gas mixture of MIC in He was prepared manometrically in a 12 L Pyrex bulb. Fourier transform
infrared spectroscopy (FTIR) measurements (1 cm$^{-1}$ resolution, 425 cm pathlength) using a previously determined
absorption spectrum from our laboratory (Papanastasiou et al., 2020) confirmed the mixing ratio, to within 10% of the
manometric preparation. Standard dilute solutions of HAN in acetone were prepared volumetrically using 1.0–2.0 μL
of the commercial HAN solution and 5.0 or 10.0 mL of acetone. The standard solutions were stored in a chemical
refrigerator in vacuum sealed Pyrex reservoirs. Samples used in the calibration infusion experiments were extracted
from the standard solution with a gas-tight 10 or 100 μL syringe.
**3. Results and Discussion**
**3.1 Methyl isocyanate (CH$_3$NCO, MIC)**
ToF-CIMS and Q-CIMS measurements with MIC mixing ratios of up to 860 ppb yielded no measurable signal for the
I-MIC$^-$ or I(H$_2$O)-MIC$^-$ adducts above the background (1σ background noise level of 3 ncps). Measurements were
performed over the range of CIMS conditions described in the experimental section. The sensitivity upper-limits
obtained are given in **Table 1**, and represent the signal increase (normalized to 1 million cps of the sum of I$^-$ + I(H$_2$O)$^-$
per mixing ratio unit increase of the compound). Our measurements indicate that MIC mixing ratios of greater than
1 ppm may be required to generate a detectable I-CIMS signal, if MIC can be detected by I-CIMS at all.
**Table 1.** ToF-CIMS and Quad-CIMS instrument sensitivity (S) and limits of detection (LOD) measured in this work
for hydroxyacetonitrile (HOCH$_2$CN, HAN) and methyl isocyanate (CH$_3$NCO, MIC) for typical field operating
conditions with an IMR temperature of 30°C, I(H$_2$O)$^-$/I$^-$ ratio of 0.55, and 1 s integration. See **Fig. 3** for sensitivity
dependence on operating conditions.

| | CH$_3$NCO (MIC) | | HOCH$_2$CN (HAN) | |
|---|---|---|---|---|
| Instrument | S | LOD | S | LOD |





|  | (ncps ppb⁻¹) | (ppb) | (ncps ppt⁻¹) | (ppt) |
|---|---|---|---|---|
| ToF-CIMS | <0.009 | >860 | 19.6 ± 0.5 | 0.3 |
| Quad-CIMS | <0.044 | >570 | 25.8 ± 0.7 | 3 |

The ToF- and Quad-CIMS instruments used in the present study were found to be insensitive to the detection of
$CH_3NCO$ (MIC). The lack of MIC sensitivity for both instruments suggests that the I⁻ cluster with MIC is not
thermodynamically stable in the IMR. The ion focusing or ion optics in the ToF-CIMS and the Quad-CIMS are,
however, quite different. The ToF-CIMS SSQ was set to 1.7 hPa, so collisions occur in this region, resulting in some
collisional dissociation. The BSQ focuses ions and can also result in fragmentation of I⁻ clusters. The ion optics of
the Quad-CIMS uses only static electric fields in a low-pressure region ($<1.3 \times 10^{-4}$ hPa) and does not have sufficient
frequency to dissociate I⁻ clusters. Given the lack of sensitivity of both ToF- and Quad-CIMS instruments to MIC, it
is unlikely that other common I-CIMS instrument configurations have the sensitivity to detect MIC at atmospherically
relevant mixing ratios.

**3.2 Hydroxyacetonitrile (HOCH₂CN, HAN)**

The diffusion source, with the commercial 70% HAN/H₂O solution, proved to be an unsuccessful HAN delivery
method. This method produced high gas-phase concentrations of an HCN impurity as identified by I-CIMS. Although
HCN was a minor sample impurity, <1%, its high vapor pressure (862 hPa at 295 K (Perry and Porter, 1926)) and low
Henry's law coefficient (~9 M Atm⁻¹ (Burkholder et al., 2019)), resulted in a [HCN]/[HAN] gas-phase mixing ratio
of greater than 1000, i.e., HAN was not detectable by I-CIMS using this source. The high gas-phase HCN
concentration, using the diffusion method, precluded quantitative calibration of HAN concentration using the total
reactive nitrogen calibration method.
The infusion source did not produce detectable gas-phase HCN above the detection limit of the mass spectrometers,
i.e., the HCN impurity level was much less than 1%. Therefore, HCN did not influence the absolute HAN calibration
determination using the Nᵣ instrument. The region around the infusion source was heated to 50°C when using the
commercially available HAN stock solution. Although this source worked, it didn't provide a stable HAN signal to
within 10%. This source, using a dilute HAN/acetone mixture with the region of the infusion source heated to 45°C,
i.e., slightly below the boiling point of acetone, yielded more stable HAN signals with variations of a few percent.
Measurements performed at temperatures greater than 45°C yielded reasonable results, but the signal was less stable
due most likely to boiling of the acetone solvent.
Calibration of hydroxyacetonitrile (HAN) solutions by total reactive nitrogen (Nᵣ) is shown in **Fig. 1**. The infusion
method produces a stable source of HAN with a high signal-to-noise ratio for a typical calibration experiment. The
total HAN concentration was set by adjusting the injection flow rate. Individual solutions were calibrated multiple
times, as shown in **Fig. 1**. The 2σ uncertainty of the fit precision of [HAN] vs. infusion flow rate was determined to
be 4%. The small positive intercept may be due to minor unidentified Nᵣ impurities.

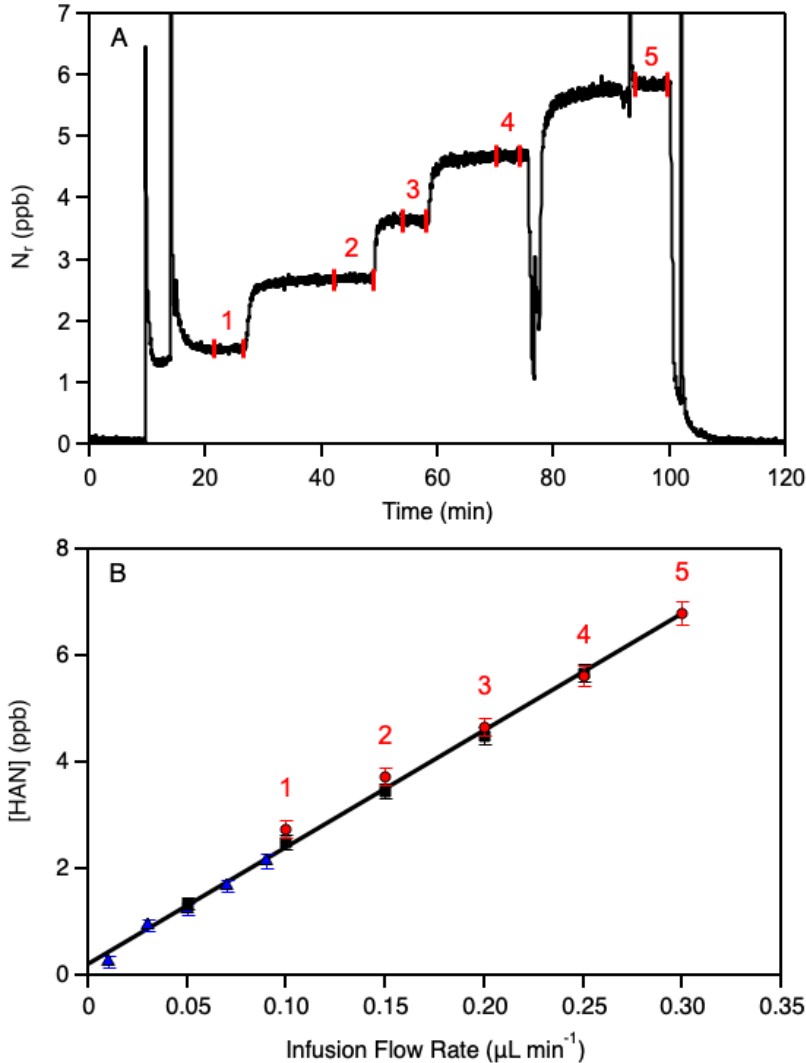

**Figure 1**: Calibration of the hydroxyacetonitrile (HOCH$_2$CN, HAN) infusion source at 23°C using the total nitrogen, N$_r$, instrument (see text for details of the N$_r$ instrument). **(a)** Background-corrected time series of a representative calibration experiment. The data within the numbered vertical lines were averaged and correspond to the numbered points in panel b. **(b)** Calibration of HAN concentration as a function of infusion source flow rate. Different symbols represent independent calibration experiments. The line is an unweighted linear least-squares fit of all the data. Error bars represent 2σ measurement precision.

Calibrations of the HAN signal on the ToF-CIMS and Quad-CIMS instruments were made by varying the infusion flow rate, see representative data for the Quad-CIMS in **Fig. 2**. The obtained ToF- and Quad-CIMS sensitivity for HAN given in **Table 1** was obtained using the infusion method with the dilute HAN/acetone samples. For typical field operating conditions with an IMR temperature of 30°C, I(H$_2$O)$^-$/I$^-$ ratio of 0.55, and 1 s integration the HAN sensitivity of the ToF I-CIMS was determined to be ~20 ncps ppt$^{-1}$.



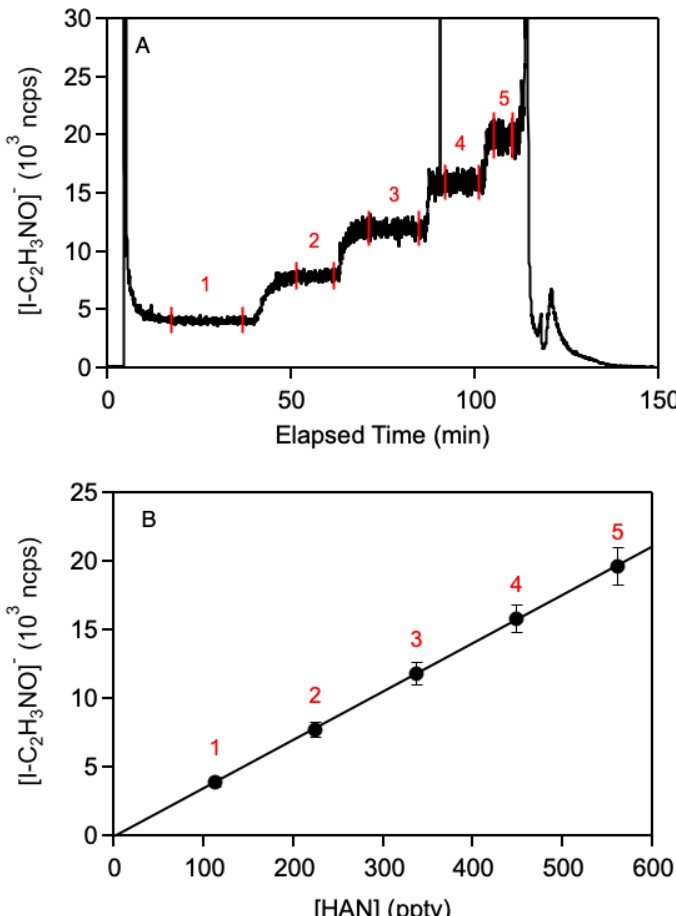

**Figure 2**: Calibration of the Quad-CIMS instrument using the hydroxyacetonitrile (HOCH$_2$CN, HAN) infusion source at 23°C (see text for details of the Quad-CIMS instrument). (a) Background-corrected time series of a representative Quad-CIMS calibration experiment (IMR temperature = 20°C, I(H$_2$O)$^-$/I$^-$ = 0.57). The data within the numbered vertical lines were averaged and correspond to the numbered points in panel B. (b) Quad-CIMS HAN calibration curve where the HAN concentration was determined from the N$_r$ calibration of the infusion source, e.g., see **Fig. 1**. The line is an unweighted linear least-squares fit of the data. Error bars represent 2σ measurement precision.

The HAN concentration from the infusion source at the instrument inlet was varied, for each IMR temperature and I(H$_2$O)$^-$/I$^-$ ratio, to determine the HAN sensitivity temperature and I(H$_2$O)$^-$/I$^-$ dependence shown in **Fig. 3**. The ToF-CIMS and Quad-CIMS instruments are both highly sensitive to HAN, but displayed slightly different temperature and I(H$_2$O)$^-$/I$^-$ dependencies. As the temperature increased, the HAN sensitivity decreased consistent with the [I-HAN]$^-$ adduct being less stable at higher temperatures. For the ToF-CIMS instrument, the HAN sensitivity decreased a factor of ~1.5–2 between 20 and 40°C at the highest I(H$_2$O)$^-$/I$^-$ ratio included in this study. For the Quad-CIMS instrument, a ~25% decrease in HAN sensitivity was observed at a I(H$_2$O)$^-$/I$^-$ ratio of 1 when increasing the IMR temperature from 10 to 30°C. The I(H$_2$O)$^-$/I$^-$ dependency for HAN sensitivity was fit reasonably well with a quadratic dependence on I(H$_2$O)$^-$/I$^-$: A + B(I(H$_2$O)$^-$/I$^-$) – C(I(H$_2$O)$^-$/I$^-$)$^2$, where the constant term represents HAN clustering with I$^-$, reaction 1; the linear term represents HAN undergoing a ligand switching reaction with I(H$_2$O)$^-$, reaction 2; and the quadratic term may represent a lack of reactivity of higher-order H$_2$O clusters, I(H$_2$O)$_n^-$ where n >1 or a shift in the





238     $(I\text{-}HAN)^- + H_2O \leftrightarrow (I\text{-}H_2O)^- + HAN$                    (3)

239     equilibrium.

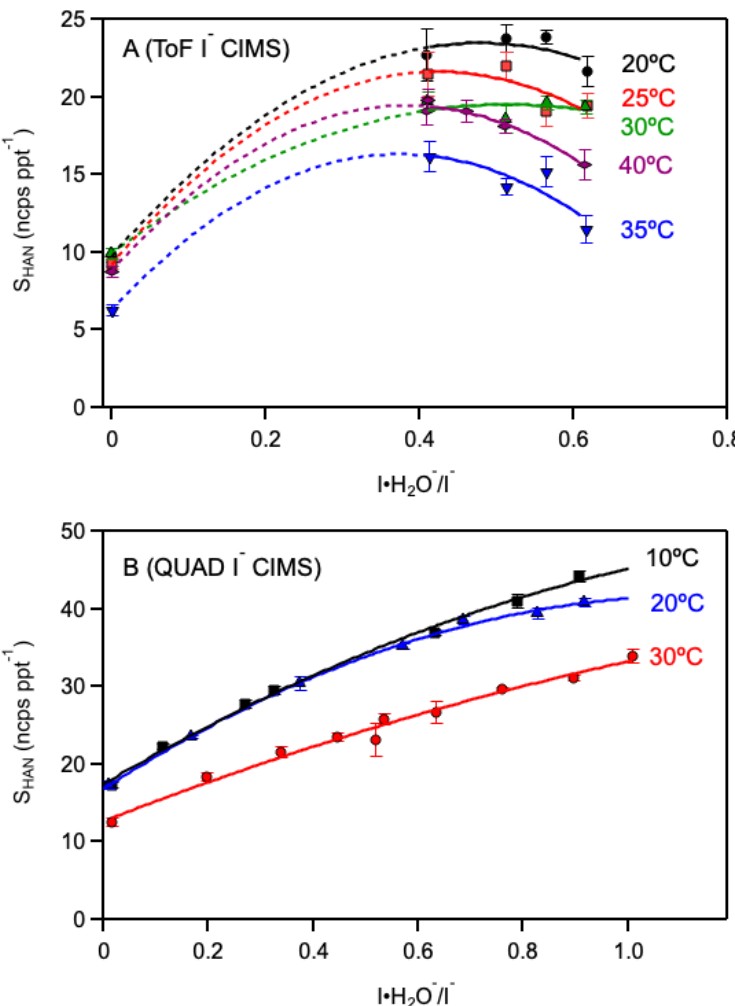

240

**Figure 3**: Hydroxyacetonitrile (HOCH$_2$CN, HAN) calibration factors for the ToF and QUAD mass spectrometers used in this work as a function of the I(H$_2$O)$^-$/I$^-$ ratio and ion-molecule reactor, IMR, temperature. Error bars represent 2σ precision of the linear calibration fits. Lines are empirical polynomial fits to guide the eye. (a) HAN calibration factors measured for the ToF-CIMS. Symbols represent IMR temperatures of 20°C (black circles), 25°C (red squares), 30°C (green triangles), 35°C (blue upside-down triangles), and 40°C (purple diamonds). (b) Quad-CIMS HAN calibration data. Symbols represent IMR temperatures of: 10°C (black squares), 20°C (blue triangles), 30°C (red circles).





There is a plausible explanation for the differences in the Quad-CIMS and ToF-CIMS instrument $I(H_2O)^-/I^-$
sensitivity dependence. First, there are differences in these instruments in the ion focusing downstream of the IMR.
The ToF-CIMS contains an SSQ at 1.73 hPa that imparts an electric field and acts as a collisional dissociation chamber.
The subsequent BSQ also imparts an electric field that can dissociate weakly bound $I^-$ clusters. The $I(H_2O)^-/I^-$ is
dynamically controlled in the ToF-CIMS by what is observed at the detector. Therefore, the $I(H_2O)^-$ (and $I(H_2O)_n^-$)
ion counts are not representative of the $I(H_2O)^-/I^-$ ratio in the IMR, i.e., the $I(H_2O)^-/I^-$ ratio in the IMR is being under-
estimated, although we were unable to quantify the dependence. The Quad-CIMS, which has ion lenses at low-
pressure downstream of the IMR, 1.3–0.13) $\times$ $10^{-3}$ hPa, should have only minor collisional dissociation, if any.
Therefore, the $I(H_2O)^-/I^-$ ratio measured by the Quad-CIMS is, most likely, close to the actual $I(H_2O)^-/I^-$ in the IMR.
In conclusion, it is recommended that instruments used to quantify HAN be calibrated under actual operating
conditions. The protocols presented in this work can be used for the calibration of HAN.
**4 Conclusions**
In this study, protocols for determining I-CIMS instrument sensitivity for $CH_3NCO$ (MIC) and $HOCH_2CN$ (HAN),
two stable toxic $C_2H_3NO$ isomers (Panda et al., 2023), were developed. Calibration of ToF-CIMS and Quad-CIMS
instruments for HAN were performed over a range of instrument conditions commonly used in laboratory and field
experiments, including: ion molecule reactor (IMR) temperature (10–40°C) and $I(H_2O)^-/I^-$ ratios between 0.1 and 1.
Both I-CIMS instruments were found to be highly sensitive to HAN with 0.3 and 3 ppt limits of detection (LOD) for
the ToF-CIMS and Quad-CIMS instruments, respectively, for measurements with the IMR at 30°C, $I(H_2O)^-/I^-$ ratio of
0.55, and 1 s integration (see **Table 1**). The instruments were insensitive to MIC with LODs of >860 and >570 ppb
for the ToF and Quad I-CIMS instruments, respectively. A weak negative temperature dependence and systematic
positive $I(H_2O)^-/I^-$ ratio dependence was observed for HAN. The ToF and Quad I-CIMS instruments have similar
normalized sensitivities, reflecting the similar chemistry in the ion molecule reaction regions. The ToF instrument
has much lower LOD because the reagent ion concentration was approximately 10 times greater than that in the Quad
instrument and the ToF has far better mass resolution. The results from this study should, to a first approximation,
translate to other laboratory and field I-CIMS instruments.
Our work demonstrates that the previous field studies of Priestley et al. (2018), Mattila et al. (2020a; 2020b), and
Wang et al. (2022), which used I-CIMS detection methods, mis-attributed the $C_2H_3NO$ mass signal as methyl
isocyanate, MIC, and also provides evidence for the observation of HAN, a previously unrecognized species in these
environments. Our results suggest that HAN was observed, but do not imply that MIC was not present in the
environments studied by Priestley et al., Mattila et al., and Wang et al. Since I-CIMS is not sensitive to MIC,
alternative measurement methods, such as proton transfer CIMS, would be required to identify the presence of MIC.
Our work indicates that HAN is commonly present in the troposphere. The heterogeneous and gas-phase atmospheric
chemistry of $HOCH_2CN$ (HAN) are, however, not presently characterized. Here, we postulate that in addition to
primary HAN emissions, e.g., from wildfires, that HAN would be formed heterogeneously in clouds, or on hydrated
aerosol, via the liquid-phase reaction:
$HCN + H_2CO \Leftrightarrow HOCH_2CN$                     (4)
The partitioning of HAN between the liquid- and gas-phase will depend on its Henry's law coefficient, which has not
been measured to date. Sander (2023) reports an estimated Henry's law coefficient value of ~130 M atm$^{-1}$, using a
quantitative structure-property relationship, which implies partitioning of HAN into the gas-phase. HCN and $H_2CO$



are ubiquitous in the atmosphere with elevated concentrations in wildfire plumes, which may lead to a significant
enhancement of the HAN concentration following a wildfire plume exposure to clouds. Our work will aid future
laboratory and field studies to identify the atmospheric source chemistry of HAN as well as its atmospheric loss
processes and degradation products.
*Data availability*. NA
*Author contributions*. ZF undertook the experimental measurements and contributed to the first draft and writing of
the paper. AC performed the MIC infrared measurements and initial ToF MIC measurements. JAN performed the
initial ToF MIC measurements. JMR performed the initial Nr measurements and contributed to the writing of the
paper. JBB supervised the project and completed the writing of the paper.
*Competing interests*. The authors declare that they have no conflict of interest.
*Financial support*. This research was supported in part by the NOAA Climate Goal and NASA Atmospheric
Composition Programs.
Review statement.



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
