# Peer review of "isocyanate (CH3NCO) Isomers using I- Chemical Ionization"

_Atmospheric Measurement Techniques, 2024_

## Author Comment (AC1)

Response to John Crounse review of:
"Calibration of Hydroxyacetonitrile (HOCH$_2$CN) and Methyl isocyanate (CH$_3$NCO) Isomers using I$^-$ Chemical Ionization Mass Spectrometry (CIMS)", Finewax, Chattopadhyay, Neuman, Roberts, and Burkholder
The authors thank the reviewer for their careful reading and constructive comments regarding our manuscript.

**Reviewer Comment:**
The authors should modify the strong, confrontational language used in abstract, introduction and conclusion where they apply results from this work to reports of previous measurements. Such statements may be hard to prove, and serve no good end. Statements including modifiers such as 'likely', 'may have', and 'it may be' should suffice to convey the authors' point.

> **Author Response:**
> The use of "contradict" in the abstract was a concern in the Farmer review as well and the text will be modified as outlined in that review response. It was not our intention to diminish the research reported in previous field studies, but to identify and clarify the error in MIC identification using I-CIMS detection. Our work is definitive regarding the identification of MIC. We are careful to say that the C2H3NO I-CIMS signal is most likely due to HAN, but we are not definitive on this point.

> **Action to be taken:**
> In addition to the revisions outlined in the Farmer review response, we will revise the conclusion text "Our work indicates that HAN is commonly present in the troposphere." As follows "Our work indicates that HAN is likely to be  present in the troposphere.".

**Reviewer Comment:**
Are there other stable isomers having formula C2H3NCO besides the two discussed in this paper? If so, authors should discuss the likelihood that these could contribute to ambient CIMS signals. Are there other 'nearby' isobars (ie, different atomic composition), considering the resolution of spectrometers in question, which could contribute to signal with nominal mass of C2H3NCO clusters?

> **Author Response:**
> We assume that the reviewer means the formula C2H3NO. There are other compounds with this chemical formula, e.g., N-methyleneformamide is a candidate. N-methyleneformamide, however, does not contain the necessary acidic or polar H that would likely be required for detection by I-CIMS. We are careful in the manuscript to say that HAN is the likely compound being detected, but we can't be definitive.

> The HR-ToF I-CIMS mass resolution of 5000 allows for near unamibiguous detection of the C2H3NO chemical formula.

> **Action to be taken:** None

**Reviewer Comment:**
The IR absorption bands and cross-sections used to quantify CH3NCO should be included. How stable was this compound in AL cylinders? How do Nr calibration of CH3NCO mixtures in AL bottles agree with FTIR determinations?

> **Author Response:**
> The infrared absorption spectrum of CH$_3$NCO (MIC) was reported in a previous study from this laboratory (Papanastasiou, Bernard, and Burkholder: Atmospheric fate of methyl isocyanate, CH$_3$NCO: OH and Cl reaction kinetics and identification of formyl isocyanate,

HC(O)NCO, Earth Space Chem., 4, 1626-1637, https://doi.org/10.1021/acsearthspacechem.0c00157, 2020.) as cited in the experimental section of our manuscript. The stability of the MIC sample prepared in aluminum cylinders was not tested. The I-CIMS response, however, was tested using both the diluted MIC samples in aluminum cylinders and using the ~3.5% mixture prepared in a 12 L Pyrex bulb and measured by FTIR. The text in the experimental section did not make this clear. $N_r$ calibration of $CH_3NCO$ samples was not part of the present study.

**Action to be taken:**
The text in section 2.1 "Methyl isocyanate standards were quantitatively added to the calibrated zero air flow sampled by the I-CIMS instruments." was revised as follows: "Methyl isocyanate bulb and aluminum cylinder standards were quantitatively added to the calibrated zero air flow sampled by the I-CIMS instruments.".

**Reviewer Comment:**
Are there other possible 'N' compounds produced in the syringe pump std method for HOCH2CN? How does the mixing ratio calculated from pumping rate and gas flow compare with Nr determination?

**Author Response:**
The most likely 'N' compound impurity in the $HOCH_2CN$ (HAN) sample was HCN. CIMS measurements using a diffusion source of the commercial HAN sample showed significant gas-phase HCN signals due to the much higher vapor pressure of HCN than HAN (explained in section 3.2). The infusion source, however, limited the gas-phase HCN concentration to below the CIMS detection level. In section 2.2, we state that the HAN concentration was not determined using pumping and flow rates. Instead the HAN concentration was determined using the $N_r$ method.

**Action to be taken:** None

**Reviewer Comment:**
Figure 1: y-axis of Panel B is Nr signal or something else? If Nr, you should keep this label, and state in text explicitly the assumption that 100% (or whatever the assumption is) Nr signal is comprised of HAN.

**Author Response:**
Panel B y-axis label, [HAN] (ppb), was made by taking the $N_r$ signal concentration to be equivalent to the HAN concentration. The figure caption can be revised to make this clear.

**Action to be taken:**
The figure caption text "Calibration of HAN concentration as a function of infusion source flow rate." Was revised as follows: "Calibration of HAN concentration, taking the $N_r$ measured concentration, example in panel A, to be equal to the HAN concentration, as a function of infusion source flow rate.".

**Reviewer Comment:**
Figure 3: How do the authors interpret and deal with the ToF 30C and 35C and Quad 20C curves that seem to be out of family with the other curves, and the mechanism. Are these curves reproducible?

**Author Response:**
Figure 3 presents results obtained in multipoint calibration measurements performed over the course of this project, i.e., several weeks. The scatter in the data plotted represents the random and systematic errors in the measurement. The data obtained at 30C for the ToF I-CIMS instrument does not appear to cleanly follow the data trend. However, this does not discredit

the dataset. We have proposed a possible interpretation for the data trend in the text following Figure 3.

**Action to be taken:** None

**Reviewer Comment:**
LN272-281: This PP should be reformulated. Suggest that if the authors wish to put forward the idea that HAN is observed in the atmosphere it would be more appropriate, straightforward, and convincing, if they present their own data, rather than simply re-assigning previously published by other groups. [reviewer notes that the instrumentation calibrated within this work has been deployed I number of previous field campaigns from aircraft and ground-based platforms, with plenty of biomass burning influence]. In addition, the authors should discuss the more general importance of $HOCH_2CN$ to the nitrile budget. What fraction of nitriles does $HOCH_2CN$ comprise? Is there reason (and if so what are the reasons) to study its chemistry in more detail?

**Author Response:**
We understand the eagerness of the reviewer for an in-depth analysis of field measurements and we agree that it is a critically important component of an evaluation of MIC and HAN emissions. A thorough study of a field campaign dataset is, however, beyond the scope of the present work, but should be addressed in future studies. It is self-evident to readers of this journal that knowing the chemistry of an atmospheric trace gas plays a critical role in understanding its impact on the environment and human health, i.e., HAN is a toxic compound.

**Action to be taken:** None

---

## Author Comment (AC2)

Response to Delphine Farmer review of:

"Calibration of Hydroxyacetonitrile (HOCH$_2$CN) and Methyl isocyanate (CH$_3$NCO) Isomers using I$^-$ Chemical Ionization Mass Spectrometry (CIMS)", Finewax, Chattopadhyay, Neuman, Roberts, and Burkholder

The authors thank the reviewer for their careful reading and constructive comments regarding our manuscript.

**Reviewer Comment:**

The reviewers concerned with wording in the following abstract sentence "These results contradict several recent field studies that have reported the detection of MIC using I-CIMS instruments.", which was interpreted to imply a controversy.

**Author Response:**

It was not our intention to imply a controversy as the results from our study are definitive in that I-CIMS instruments are not sensitive to MIC and, therefore, previous studies have mis-attributed the observed C2H3NO I-CIMS signal to MIC.

**Action to be taken:**

Revise the abstract text "These results contradict several recent field studies that have reported the detection of MIC using I-CIMS instruments. This study demonstrates that HAN, rather than MIC, was most likely the C$_2$H$_3$NO isomer observed in those field studies, although the source chemistry for HAN remains uncharacterized."

as follows:

" The present results  show that several recent field studies  using I-CIMS instrument detection have misattributed the C$_2$H$_3$NO signal to MIC. This study  proposes that HAN, rather than MIC, was most likely the C$_2$H$_3$NO isomer observed in those field studies, although the source chemistry for HAN remains uncharacterized. This study demonstrates the importance of applying absolute calibration standards in the identification and quantification of isomeric compounds.".

**Reviewer Comment:**

I encourage the authors to consider this question - what makes I-CIMS so sensitive to hydroxyacetonitrile, and not to methyl isocyanate?

**Author Response:**

We did not provide an explanation for the difference in sensitivity between the MIC and HAN isomers in our original submission. Reviewer #2 has provided an explanation in their review, which we agree with. We will add text and citation to two references that addresses this point

**Action to be taken:**

We have added text and citation to two references in the conclusion section as follows: Iyer et al, (2016) and Hyttinen et al, (2018) provide an explanation for the significant difference in the I-CIMS sensitivity for MIC (CH$_3$NCO) and HAN (HOCH$_2$CN), due to the stability of I- cluster binding energies. That is, the H-bonding with the HO group in HAN leads to a stable I- cluster, while MIC would not form a stable I- cluster.

Hyttinen, N., Otkjaer, R. V., Iyer, S., Kjaergaard, H. G., Rissanen, M. P., Wennberg, P. O., and Kurten, T.: Computational comparison of different reagent ions in the chemical ionization of oxidized multifunctional compounds, J. Phys. Chem. A., 122, 269-279, https://doi.org/10.1021/acs.jpca.7b10015, 2018.

Iyer, S., Lopez-Hilfiker, F., Lee, B. H., Thornton, J. A., and Kurten, T.: Modeling the detection of organic and inorganic compounds using iodide-based chemical ionization, J. Phys. Chem. A, 120, 576-587, https://doi.org/10.1021/acs.jpca.5b09837, 2016.

---

## Author Response (AR2)

The editors three correction have been made in the revised manuscript.

There are three minor technical corrections that I ask the authors to consider for the final manuscript:

line 95: Is there a "calibrated" zero air flow before the respective standards are added? Better reformulate or just drop "calibrated".
    Action: calibrated was deleted from the text

line 220: When directly referring to plot axes scaled in ppb I recommend using the correct term "mixing ratio".
    Action: "concentration" has been replaced by "mixing ratio" within the figure caption.

line 288: When using "stability" this should refer to compounds or ion-clusters but not binding energies. Please reformulate.
    Action: "stability" was deleted from this sentence